# Dietary variability and micronutrient status of individuals with Yaws infection in Ghana: A case-control study

Abigail Agbanyo[1,2], Michael Ntiamoah Oppong[1], Ruth Dede Tuwor[1], Pius Takyi[1], Felix Wireko[1], Philemon Boasiako Antwi[1], Dzifa Kofi Ahiatrogah[1], Aloysius Dzigbordi Loglo[1], Bernadette Agbavor[1], Alex Owusu-Ofori[2,3], Richard Odame Phillips[1,2,3], Yaw Ampem Amoako[1,2,3]*

1 Kumasi Centre for Collaborative Research in Tropical Medicine, Kwame Nkrumah University of Science and Technology, Kumasi, Ghana, 2 School of Medical Sciences, Kwame Nkrumah University of Science and Technology, Kumasi, Ghana, 3 Komfo Anokye Teaching Hospital, Kumasi, Ghana

* yamoako2002@yahoo.co.uk

## Abstract

### Background

Yaws is a neglected tropical skin disease mainly affecting children under 15 years. It is targeted for eradication by 2030 through Mass Drug Administration (MDA) of a single oral dose of azithromycin (30 mg/kg), which has achieved about a 95% cure rate. Despite this, the disease persists in endemic countries. Eradication requires addressing contributing factors, including the role of nutrition in yaws; yet this remains under-explored. This study assessed the nutritional status of individuals in yaws-endemic areas using a case-control design.

### Methods

A case-control study was conducted from May to November 2024 in 33 communities across two districts in Ghana. Cases were Dual Path Platform (DPP) positive individuals, matched by age and sex with healthy controls. Data collection included demographic and anthropometric measurements and a Ghana-specific multi-pass 24-hour dietary recall.

### Results

A total of 64 cases and 64 matched controls [median age 11.5 years, Interquartile range (IQR) 9–13.8] were enrolled. Moderate stunting was found in 27% of cases and 22% of controls; severe stunting in 6% and 5%, respectively. Roots and tubers were consumed by all participants, while fruit intake was low (5% overall; 3% in cases, 8% in controls). Controls generally had slightly higher micronutrient intake than cases, except for energy (1754 ± 657 kcal vs. 1726 ± 707.3 kcal), fat (55.15 ± 28.04 g vs.

**Data availability statement:** All relevant data are within the paper and its Supporting information files.

**Funding:** The author(s) received no specific funding for this work.

**Competing interests:** The authors have declared that no competing interests exist.

51.83 ± 27.04 g), and vitamin C (108.7 ± 42.24 mg vs. 107.7 ± 53.18 mg). No statistically significant differences in nutrient adequacy were found between healed and non-healed yaws cases (p > 0.05), though participants with non-healed lesions more often had inadequate intakes of energy (88% vs. 80%), fibre (100% vs. 96%), iron (63% vs. 54%), zinc (75% vs. 71%), and vitamin B12 (63% vs. 55%).

## Conclusion

In these yaws-endemic districts, we observed high levels of undernutrition and micronutrient deficiencies among both cases and controls. Although nutritional status was not independently linked to poorer treatment outcomes, the burden of malnutrition underscores the need for integrated health interventions. Further research is warranted to clarify the relationship between chronic nutritional deficiencies and yaws outcomes.

## Background

Yaws, a chronic bacterial neglected tropical disease (NTD), is caused by *Treponema pallidum* subsp *pertenue*. It mainly affects the skin, bones, and cartilage of its victims in poor rural areas across 16 countries, especially in West Africa, Southeast Asia, and the Pacific regions [1]. Around 75%−80% of those affected are children under 15 years old.

Clinically, yaws manifests in four main stages: primary, secondary, latent, and tertiary. The earliest and most infectious forms of the disease are wart-like tumours called papillomas, which subsequently develop into an ulcer [1]. This stage of the disease can progress to secondary yaws within months to two years, marked by extensive papillomatous or ulcerative lesions, hyperkeratotic lesions on the palms and soles, and, occasionally, bone involvement that can result in dactylitis or pain and swelling in long bones [2]. If untreated, individuals enter a latent stage during which they exhibit no clinical signs, although serological tests remain positive. Tertiary yaws, which is rarely seen today, used to occur in about 10% of untreated cases [2,3] and is marked by non-infectious and disfiguring lesions [1,4,5].

Serology is the mainstay of diagnosis, and the World Health Organization (WHO) recommends the use of a dual treponemal and non-treponemal rapid test, the Dual Path Platform Syphilis Screen and Confirm assay (Chembio Diagnostics, USA).

The disease has been targeted for eradication by 2030 [6] and currently, the recommended treatment is a single dose oral azithromycin at a dose of 30 mg/kg (maximum 2 g). A single dose of intramuscular Benzathine penicillin at 0.6 million units (maximum 1.2 million units) is recommended for cases of suspected clinical treatment failure after azithromycin, or for patients who are unable to receive azithromycin treatment [7]. The treatment of cases and their contacts through mass drug administration (MDA) has been successful, with about a 95% cure rate [8]. However, there is still ongoing transmission and a great burden of the disease in endemic areas [9] with reports of persistent lesions even after treatment [10]. There is therefore an urgent

need to identify factors that could undermine the eradication efforts [11,12]. While potential contributing factors such as access to azithromycin, azithromycin resistance [7,13], coinfections [14–16], are under investigation and consideration, the role of nutrition remains unexplored.

According to the WHO, poverty, low socio-economic conditions, and poor personal hygiene facilitate the spread of yaws [7]. In rural Ghana, where yaws remains endemic, 64.6% of the population is multidimensionally poor [17], and many children in these areas face malnutrition, with some still affected by wasting, the deadliest form of malnutrition [18].

There are reported associations between nutritional inadequacies and some NTDs, such as leprosy [19,20], soil transmitted helminthiasis [21], schistosomiasis [22], leishmaniasis [23] and recently, Buruli ulcer [24,25]. In 2024, a systematic review and meta-analysis identified an association between malnutrition and NTDs, revealing consistent deficiencies in nutrients such as iron, selenium, zinc, and vitamin A [26].

A report examining Brazil's national effort to eliminate yaws from 1956–1961 using single-dose penicillin injections revealed widespread malnutrition and starvation among individuals affected by yaws [27], prompting discussions on integrating nutritional support into disease control and elimination strategies. Drawing lessons from India's successful eradication of yaws, their efforts highlighted the detrimental effect that yaws infection has on the nutritional status of its affected population, a critical lesson for informing global eradication strategies [28]. However, there is a paucity of data on the nutritional status of yaws-endemic communities, including those in Ghana, which hinders understanding of its role in infection, disease progression, and treatment outcomes. This study aimed to investigate the nutritional status of individuals in yaws-endemic regions using a case-control approach. Specifically, we investigated the nutritional status of children with yaws in two Ghanaian districts, exploring its possible role in infection and treatment outcomes to inform policy and guide integrated strategies to support yaws eradication efforts.

## Methods

### Ethical statement

The protocol for the study was reviewed and approved by the Committee on Human Research, Publications, and Ethics of the School of Medical Sciences at the Kwame Nkrumah University of Science and Technology, Ghana (approval number: CHRPE/AP/361/24). The content of the participant information leaflet was explained to the participants and/or their legal representative, mostly in Twi. Written informed consent was obtained from all participants. For participants <18 years old, consent was obtained from a parent or a legal guardian. Additionally, assent was obtained from participants aged 12–17 years. The study was conducted in accordance with the principles outlined in the Declaration of Helsinki [29].

### Study setting

The study was carried out across 33 communities in two districts of Ghana: 29 communities in Wassa Amenfi East (Western Region) and 4 communities in Aowin (Western North Region). The districts (Fig 1) are predominantly rural, with about 77% of the population in Wassa Amenfi East and 87% in Aowin living in rural communities. The primary economic activities in these districts are agriculture, mainly involving the cultivation of crops such as oil palm, cocoa, rice, cassava, maize, plantain, and cocoyam [30,31], with a substantial contribution from mining activities, including small-scale mining [32]. Based on national surveillance data, these districts are situated in regions that have consistently reported yaws cases over recent years [33].

### Study design, population, and sampling

This was a case-control study, and participants were recruited and followed up from 04/06/2024 to 05/12/2024. The study population consisted of individuals who were confirmed positive for yaws using the Dual Path Platform (DPP) Syphilis screen and confirm assay (Chembio Diagnostic Systems, Medford, NY, USA). These individuals were recruited as part

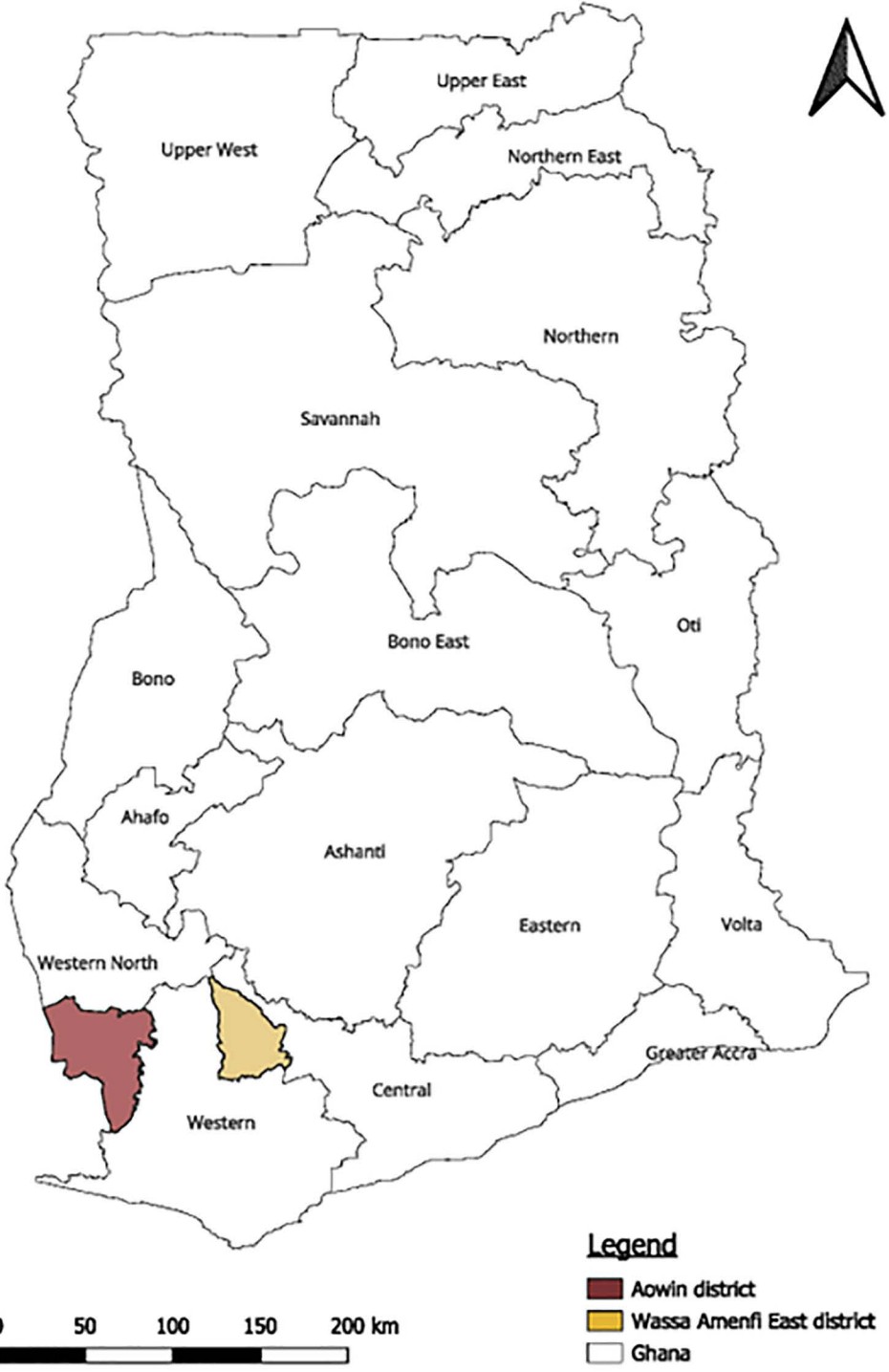

**Fig 1. Map of Ghana showing the the two study districts.** Map was generated using ArcGIS 10.7.1 (Esri Inc., Redlands, California, USA). The shapefiles for Ghana and the various regions obtained from OpenStreetMap (https://www.openstreetmap.org/copyright, CC BY-SA 2.0) were utilized as data sources for plotting the map. Map data from OpenStreetMap https://www.openstreetmap.org/copyright.

of a larger study that involved active case-finding activities for yaws in the two districts. The lesion type of confirmed yaws cases was assigned by experienced clinicians based on the WHO Yaws Recognition Booklet for Communities [34]. All positive cases were treated with a single oral dose of 30 mg/kg of azithromycin as recommended [12], administered by trained local health nurses at baseline. Using the assistance of school teachers, age- and sex-matched controls (as closely as possible) were identified and recruited from the same community for each included yaws case. Age and sex matched controls were selected from within the same household as cases. Where no age and sex matched controls were available from within the same household, individuals living in the same community but who had negative DPP test results and did not have any clinical evidence of yaws were selected as controls. Health personnel and researchers participating in the study underwent a one-day training session on the use of the data collection tools, the types of data to be gathered, and the appropriate methods for data collection before the start of the study. Due to the focal endemicity and uneven distribution of yaws as a disease, no formal sample size calculation was done, and the sample size was based on the convenience of matching cases identified during the active case search to controls.

### Baseline demographic and clinical assessment

Following informed consent, demographic information for all participants (cases and controls) and clinical information (cases only) were collected on the WHO Skin NTDs clinical and treatment form [35]. We then assessed dietary intake using the 24-hour recall questionnaire as well as anthropometric measurements in cases and controls.

### Anthropometric measurements

Following the World Health Organization's protocol for measuring children [36], anthropometric data such as weight and height, were obtained from cases and controls. Weight measurement was done using the Omron HN286 electronic human weighing scale (Omron Healthcare UK Ltd, Milton Keynes, MK, United Kingdom). Participants removed their shoes, emptied their pockets, and stood still alone on the scale, which was positioned on a stable, flat surface and set to zero. The weight was recorded to the nearest 0.1 kg.

Height was measured with a Seca 213 mobile stadiometer (Seca, Hamburg, Germany), placed on a firm surface against the wall. Participants stood on the board with feet together, heels against the backboard, and knees straight. While looking straight ahead with eyes level with the ears, the height was read to the nearest 0.1 cm, with the measuring arm gently on the head. BMI was calculated from weight and height with the formula: BMI (kg/m$^2$) = weight (kg)/ height$^2$ (m$^2$). Classification for thinness and stunting followed the WHO 2006 growth standards for 5−19 years [37,38] Using the WHO AnthroPlus (WHO; Version 1.0.4) [39] Z scores for weight-for-age (WAZ) and height-for-age (HAZ) were calculated to assess undernutrition (thinness and stunting respectively) [40]. The calculated sex appropriate BMI-for-age were categorized as; 'severe thinness' (Z-score < −3 SD), 'thinness' (Z score < −2 SD), 'normal' (−2 ≤ Z-score ≤ +1 SD), 'overweight' (Z score > +1 SD), and 'obese' (Z-score > +2 SD). [40]. The calculated sex appropriate height for age were categorized as: 'severely stunted' (Z-score < −3), 'moderately stunted' (Z-score ≥ −3 to < −2), and 'not stunted' or normal (Z-score ≥ −2) [41,42].

### Dietary assessment

A Ghana-specific multi-pass 24-hour recall dietary questionnaire [43] was administered to participants. Data on food consumption across two weekdays and one weekend day were collected from participants starting from the previous day before our interview. This open recall method required participants to recall all foods, beverages, and snacks consumed each day on two weekdays and a weekend, inside and outside their homes. Participants were prompted about drinks and snacks for the purpose of accuracy. The 24-hour recall questionnaire was administered face-to-face on weekdays, and no interviews were conducted on weekends or public holidays. Quantification of recorded consumed food was done using household measures and food models. This was subsequently converted into grams using a standardized unit of measure

from a nutrient analysis template for each participant; furthermore, the nutritional contents of recorded foods were also estimated [44]. Each participant's nutrient intake was compared to the relevant cut-off for their age (and sex if relevant), using information from the Estimated Average Requirement (EAR) of the Food and Agriculture Organization (FAO)/WHO [45] or the Recommended Dietary Allowance (RDA) [46] if no EAR was available.

Proportions of participants' dietary diversity were calculated from data collected through the multi-pass 24-hour recall for each participant. Food groups were categorized into nine major food groups based on modification of the USAID indicator guide and the Ministry of Food and Agriculture, Ghana Food Based Dietary Guidelines (FBDG) [47,48]. These food groups are: (i) grains, roots, and tubers; (ii) green leafy vegetables; (iii) flesh foods (meat, fish, poultry and liver/organ meats); (iv) legumes, pulses and nuts; (v) eggs; (vi) fruits; (vii) dairy products (milk, yogurt, cheese); (viii) other vegetables; (ix) oil seeds.

### Follow-up

Cases were evaluated four weeks after treatment to determine the effectiveness of yaws therapy. Lesions were labelled as healed or not healed. A lesion was considered healed if it showed complete re-epithelialization or if the lesion had resolved without residual exudate, crust, or open wound, and the nearby skin appeared normal under natural light. Conversely, a lesion was marked as not healed if any open wound, exudate, crust, or ongoing erythema or induration was present, as judged by physicians experienced in the clinical assessment of yaws. Additionally, anthropometric measurements and a 24-hour recall questionnaire were conducted with both cases and controls during the week 4 follow-up. In the current study, the baseline anthropometric measurements were used in the analysis of anthropometric characteristics.

### Data analysis

Data was analyzed using GraphPad Prism version 9.4.1 (GraphPad Software Inc., USA). Categorical data for cases and controls were analyzed using frequencies and proportions, with proportions expressed as a percentage of the total number in the category. When there were more than two variables, Chi-square or Fisher's exact tests were used to compare the frequencies of categorical variables. A Kolmogorov-Smirnov test for normality was performed on all continuous variables to check if they met the parametric assumptions. A Mann-Whitney U test was used to compare the medians of two groups in non-Gaussian data. Multiple comparisons employed Dunn's post-hoc correction in conjunction with the Kruskal-Wallis test. Since numerous nutrients correlated with one another, multiple t-tests were employed to compare the nutrient consumption of cases and controls.

## Results

While 71 participants tested positive on the DPP test (and were identified as yaws cases), this study presents results for 64 participants with complete data. Seven participants were excluded from the analysis due to incomplete anthropometric and/or 24-hour recall data.

### Demographic and anthropometric characteristics of study participants

The study included 128 participants, with 64 yaws cases and 64 controls. The median (IQR) age was equal for both groups at 11.5 years (9–13.8), with no significant age difference (p > 0.99) (Table 1). Males made up 74% (95/128) of the participants, and females 26% (33/128); there was no statistically significant difference (p > 0.99) in sex distribution between yaws cases and controls.

The anthropometric analysis of BMI-for-age classifications based on WHO Z-scores for children and adolescents aged 5–19 years revealed distinct patterns between the cases and control groups at baseline. However, these differences were not statistically significant, p = 0.12. Among cases, 9 individuals (14%) were identified with severe thinness, 14 (22%) had thinness, and 41 (64%) were normal, with no individuals in the overweight or obese categories. In contrast, the control

**Table 1. Demographic and anthropometric characteristics of study participants at baseline.**

| Variables | Yaws cases n=64 (%) | Controls n=64 (%) | p-value |
|---|---|---|---|
| **Age range (years)** | | | |
| *Median (IQR)* | 11.5 (9-13.8) | 11.5 (9-13.8) | >0.99[a] |
| **Sex, n (%)** | | | |
| *Male* | 47 (73) | 48 (75) | >0.99[b] |
| *Female* | 17 (27) | 16 (25) | |
| **Yaws clinical forms, n (%)** | | | |
| *Ulcer* | 42 (66) | – | |
| *Plantar* | 6 (9) | – | |
| *Papilloma* | 3 (5) | – | |
| *Squamous macules* | 11 (17) | – | |
| *Squamous macules/ulcer* | 1(1.5) | – | |
| *No lesion* | 1(1.5) | – | |
| **Anthropometric measure [c], n (%)** | | | |
| **BMI for age** | | | |
| Severe thinness | 9 (14) | 4 (6) | 0.12[b] |
| Thinness | 14 (22) | 8 (12) | |
| Normal | 41 (64) | 50 (78) | |
| Overweight | 0 | 1 (2) | |
| Obese | 0 | 1 (2) | |
| **Stunting Status [d], n (%)** | | | |
| *Not stunted* | 43 (67) | 47 (73) | 0.74[b] |
| *Moderately stunted* | 17 (27) | 14 (22) | |
| *Severely stunted* | 4 (6) | 3 (5) | |

*Counts of demographic and clinical characteristics of study participants (p-value<0.05) were considered statistically significant. [a]Mann-Whitney U tests; [b]Fisher's Exact Test; [c]Anthropometric measure used BMI-for-age for study participants aged 5–19 [37]; [d]Stunting status was based on the z-score of the WHO AnthroPlus tool. Abbreviations: IQR; Interquartile range; BMI; Body Mass Index.*

group had 4 individuals (6%) with severe thinness, 8 (12%) had thinness, 50 (78%) were normal, 1 (2%) was overweight, and 1 (2%) was obese. These distributions highlight a higher prevalence of undernutrition (as evidenced by severe thinness and thinness) among cases (23/64; 36%) compared to the control group (12/64; 19%). Conversely, the control group showed a greater proportion of individuals in the 'normal' category (50/64; 78%) and the presence of overweight (1/64; 2%) and obese (1/64; 2%). None of the individuals with yaws were overweight or obese (Table 1).

Using the WHO height-for-age z-score reference for children aged 5–19 years, a higher proportion (73%; n=47) of controls were not stunted compared to 67% (n=43) of cases. Conversely, 27% (n=17) of cases and 22% (n=14) of controls were moderately stunted. Severely stunted individuals comprised 6% (n=4) of cases and 5% (n=3) of controls. Overall, stunting was more common among cases than controls, but the difference was not statistically significant (p=0.74) (Table 1).

## Nutrient intake and diversity

The average daily intake of macro- and micronutrients for all participants was evaluated using a multi-pass 24-hour recall questionnaire. Although not statistically significant, micronutrient levels were generally lower in cases compared to controls, with controls showing slightly higher intake for most nutrients, except for energy (cases, 1754±657;

controls, 1726±707.3), fat (cases, 55.15±28.04; controls, 51.83±27.04), and vitamin C (cases, 108.7±42.24; controls, 107.7±53.18) (Table 2). The macronutrient composition revealed that carbohydrate intake was highest, accounting for 62% of total energy in the entire study population, with a mean intake of 269.6 g (Table 3). Specifically, cases consumed 62% (267.7 g) of their energy from carbohydrates, while controls also derived 62% (271.4 g). Proteins constituted the smallest portion, making up 10% of intake (42.95 g) for cases and slightly higher at 11% (47.01 g) for controls. Fat intake was marginally higher in cases (28%, 55.15 g) compared to controls (27%, 51.83 g) (Table 3).

Based on the multi-pass 24-hour recall, grain, roots, and tubers were the most consumed food items reported by 100% of participants. Legumes, pulses, and nuts were consumed by 89% of the study participants. Additionally, 18% of individuals consumed dairy products (14% among cases and 22% among controls), and flesh foods were consumed by 75% (n=48) of cases and 78% (n=50) of controls. However, only 5% of the participants ate fruits (3% among cases and 8% among controls) (Table 4). There were no sex specific differences in food group intake among participants (S1 Table).

**Table 2. Comparison of nutrient intake between study participants.**

| Variables | All | Cases | Controls | p-value |
|---|---|---|---|---|
| | n=128 | n=64 | n=64 | |
| | *Mean± SD* | *Mean± SD* | *Mean± SD* | |
| Energy (Kcal) | 1749±680.9 | 1754±657 | 1726±707.3 | 0.34 |
| Carbohydrate (g) | 269.6±99.1 | 267.7±92.8 | 271.4±105.6 | 0.80 |
| Protein (g) | 44.9±20.6 | 42.9±19.6 | 47.0±21.5 | 0.39 |
| Fat (g) | 53.5±27.5 | 55.1±28.0 | 51.8±27.0 | 0.22 |
| Fibre (g) | 19.9±8.3 | 19.8±6.9 | 20.0±9.5 | 0.94 |
| Iron (mg) | 9.3±4.3 | 9.2±3.9 | 9.4±4.8 | 0.69 |
| Selenium (µg) | 59.9±29.0 | 55.9±25.5 | 63.9±31.9 | 0.16 |
| Magnesium (µg) | 277.0±115.5 | 272.6±100.6 | 281.4±16.2 | 0.93 |
| Folic acid (µg) | 5.5±14.1 | 5.1±13.7 | 5.6±14.6 | 0.55 |
| Vitamin $B_6$ (µg) | 1.5±0.6 | 1.2±0.5 | 1.5±0.7 | 0.94 |
| Vitamin $B_{12}$ (µg) | 2.0±1.7 | 1.9±1.4 | 2.2±1.9 | 0.78 |
| Calcium (mg) | 223.1±127.7 | 210.3±107.0 | 236.0±145.2 | 0.38 |
| Vitamin C (mg) | 108.2±47.8 | 108.7±42.2 | 107.7±53.2 | 0.40 |
| Zinc (mg) | 6.6±3.1 | 6.4±2.9 | 6.8±3.2 | 0.73 |

*Continuous data are presented as means with standard deviation (SD). Continuous data were compared between cases and controls using an unpaired t-test using GraphPad Prism version 8. P-value <0.05 indicates statistical significance.*

**Table 3. Mean percentage of macronutrients in the diet of the participants.**

| Nutrients | Mean intake of nutrients | | |
|---|---|---|---|
| | All | Cases | Controls |
| Energy (Kcal) | 1749 (100) | 1754 (100) | 1726 (100) |
| Carbohydrate (g) | 269.6 (62) | 267.7 (62) | 271.4 (62) |
| Protein (g) | 44.9 (10) | 42.9 (10) | 47.0 (11) |
| Fats (g) | 53.5 (28) | 55.1 (28) | 51.8 (27) |

*Unit conversions: 1g of carbohydrate is equivalent to 4 Kcal, 1g of protein is equivalent to 4 Kcal, and 1g of fat is equivalent to 9 Kcal, which were then used to calculate percentages.*

 

**Table 4. Distribution of intake diversity among participants.**

| Food Group | Food group intake | | |
| --- | --- | --- | --- |
| | All,<br>N = 128 (%) | Cases<br>N = 64 (%) | Controls<br>N = 64 (%) |
| Grains, roots, and tubers | 128 (100) | 64 (100) | 64 (100) |
| Green leafy vegetables | 45 (35) | 22 (34) | 23 (36) |
| Flesh foods | 98 (77) | 48 (75) | 50 (78) |
| Legumes, pulses, and nuts | 114 (89) | 59 (92) | 55 (86) |
| Eggs | 29 (23) | 14 (22) | 15 (12) |
| Fruits | 7 (5) | 2 (3) | 5 (8) |
| Dairy products | 23 (18) | 9 (14) | 14 (22) |
| Other vegetables | 10 (8) | 2 (3) | 8 (13) |
| Oil seeds | 84 (66) | 43 (67) | 41 (64) |

*Food groups were categorized into nine major food groups based on modification of the USAID indicator guide and the Ministry of Food and Agriculture, Ghana Food Based Dietary Guidelines (FBDG) (Swindale and Bilinsky, 2006; Ministry of Food and Agriculture and University of Ghana School of Public Health, 2023). These food groups are: (i) grains, roots, and tubers; (ii) green leafy vegetables; (iii) flesh foods (meat, fish, poultry and liver/organ meats); (iv) legumes, pulses and nuts; (v) eggs; (vi) fruits; (vii) dairy products (milk, yogurt, cheese); (viii) other vegetables; (ix) oil seeds.*

### Adequacy versus inadequacy of nutrient intake for cases and controls at baseline

In Table 5, we evaluated the adequacy or inadequacy of the nutrients consumed by the participants using the EAR or RDA (for energy and fibre) guidelines. Both guidelines account for factors such as age and sex. A substantial proportion, 83% (n = 106), of the entire study population showed inadequate energy intake. This profound energy deficit was consistent across both cases (81%, n = 52) and controls (84%, n = 54). Regarding carbohydrate intake, most study participants (97%, n = 124) had adequate intake. This high level of adequacy was consistent across both the cases (97%, n = 62) and controls (97%, n = 62), $p > 0.99$. 68% (n = 87) of participants had adequate protein intake [67% (n = 43) of cases versus 69% (n = 44) of controls, $p > 0.99$]. Conversely, 32% (n = 41) of all participants demonstrated inadequate protein intake.

A very high proportion (92%, n = 118) of the study population had inadequate intake of fibre. This inadequacy appeared more pronounced in cases (97%, n = 62) compared to controls (88%, n = 56), $p = 0.1$. In contrast, a small fraction (8%, n = 10) demonstrated adequate fibre intake.

Furthermore, iron intake was found to be inadequate in 54% (n = 69) of the total study population. This inadequacy was evenly distributed between yaws cases (55%, n = 35) and controls (53%, n = 34). Similarly, zinc intake was inadequate in 73% (n = 93) of the study population, with no significant difference between yaws cases (72%, n = 46) and controls (73%, n = 47), $p > 0.99$.

Selenium intake was adequate in 80% (n = 102) of the study population, with similar rates in yaws cases (78%, n = 51) and controls (77%, n = 51). Vitamin B12 intake was inadequate in 56% (n = 71), with no statistically significant difference between yaws cases (56%, n = 36) and controls (55%, n = 35). Vitamin B6 intake was sufficient for 88% (n = 112) of participants, with 91% (n = 58) of cases and 84% (n = 54) of controls meeting the requirements. The rest of the study population showed inadequate intake.

Furthermore, iron intake was found to be inadequate in 54% (n = 69) of the total study population. This inadequacy was almost evenly distributed between yaws cases (55%, n = 35) and controls (53%, n = 34). Similarly, zinc intake was inadequate in 73% (n = 93) of the study population, with no significant difference between yaws cases (72%, n = 46) and controls (73%, n = 47), $p > 0.99$.

Interestingly, for both calcium and folic acid, a stark and universal inadequacy was observed, with 100% (n = 128) of the entire study population having inadequate intake. This complete deficiency was consistently present in both yaws cases

**Table 5. Adequacy of nutrient intake among participants.**

| Variables | Frequency | All | Cases | Controls | p-value |
|---|---|---|---|---|---|
| | | n = 128 (%) | n = 64 (%) | n = 64 (%) | |
| Energy (Kcal) | Adequate | 22 (17) | 12(19) | 10 (16) | 0.82 |
| | Inadequate | 106 (83) | 52(81) | 54 (84) | |
| Carbohydrate (g) | Adequate | 124 (97) | 62 (97) | 62 (97) | >0.99 |
| | Inadequate | 4 (3) | 2 (3) | 2 (3) | |
| Protein (g) | Adequate | 87 (68) | 43(67) | 44 (69) | >0.99 |
| | Inadequate | 41 (32) | 21(33) | 20 (31) | |
| Fibre (g) | Adequate | 10 (8) | 2(3) | 8(12) | 0.10 |
| | Inadequate | 118 (92) | 62(97) | 56(88) | |
| Iron (mg) | Adequate | 59 (46) | 29(45) | 30(47) | >0.99 |
| | Inadequate | 69 (54) | 35(55) | 34(53) | |
| Zinc (mg) | Adequate | 35 (27) | 18 (28) | 17(27) | >0.99 |
| | Inadequate | 93 (73) | 46(72) | 47(73) | |
| Selenium (µg) | Adequate | 102 (80) | 51 (80) | 51 (80) | >0.99 |
| | Inadequate | 26 (20) | 13 (20) | 13 (20) | |
| Vitamin B12 (µg) | Adequate | 57 (44) | 28 (44) | 29 (45) | >0.99 |
| | Inadequate | 71 (56) | 36 (56) | 35 (55) | |
| Vitamin C (mg) | Adequate | 122 (95) | 62 (97) | 60 (94) | 0.68 |
| | Inadequate | 6 (5) | 2 (3) | 4 (6) | |
| Vitamin B6 (µg) | Adequate | 112 (88) | 58 (91) | 54 (84) | >0.99 |
| | Inadequate | 16 (5) | 6 (9) | 10 (16) | |
| Magnesium (mg) | Adequate | 79 (62) | 40 (63) | 39 (61) | >0.99 |
| | Inadequate | 49 (38) | 24 (37) | 25 (39) | |
| Calcium(mg) | Adequate | 0 | 0 | 0 | >0.99 |
| | Inadequate | 128 (100) | 64(100) | 64 (100) | |
| Folic acid (µg) | Adequate | 0 | 0 | 0 | >0.99 |
| | Inadequate | 128 (100) | 64(100) | 64 (100) | |

*Proportions of cases and controls with adequate and inadequate energy and nutrient intakes were compared using a two-tailed Fisher's exact test. Categorical data are presented as percentages. Fisher's exact p-value; Estimated Average Requirement (EAR) cut-offs were used for Carbohydrate, Protein, Iron, Zinc, Selenium, Vitamin B12, Vitamin B6, Vitamin C, Magnesium, Calcium, and Folic acid. Recommended Dietary Allowance (RDA) cut-off was used for Energy, Fibre, and Vitamin K. µg/day: Microgram per day, mg/day: Milligram per day. Mean nutrient intake was categorized as adequate or inadequate based on the EAR and sometimes RDA using WHO/FAO values. The EAR and RDA cutoffs are based on age and sex [49].*

(n = 64) and controls (n = 64) for both nutrients. There were no sex specific differences in nutrient adequacy among participants (S2 Table).

## Comparison of nutrient intake of cases and controls

Fig 2 illustrates the comparative median nutrient intake trends for cases and controls at baseline. Cases had modestly higher median intakes of energy, fat, fibre, vitamin C, vitamin B6, vitamin B12, and magnesium, while controls showed slightly higher median intakes of carbohydrate, protein, zinc, iron, calcium, folic acid, and selenium.

At follow-up, there was a noticeable shift in nutrient intake between cases and controls. Cases exhibited higher nutrient intake for carbohydrate, fat, fibre, vitamin B6, and magnesium. Controls, on the other hand, had higher nutrient intake for energy, protein, iron, zinc, calcium, selenium, vitamins C and B12. Folic acid was comparable between the groups, and there was statistical significance observed in the mean intake of calcium (p = 0.02) and selenium (p = 0.03) (Fig 3).

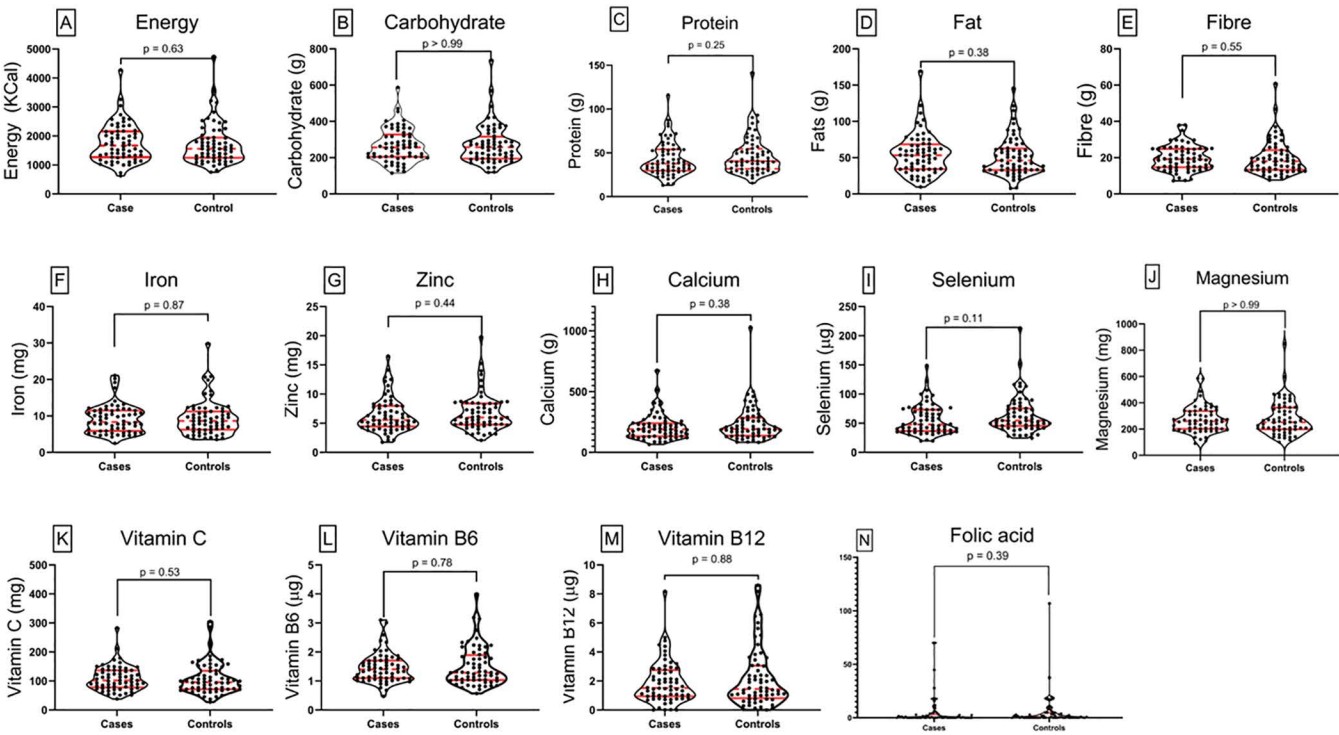

**Fig 2. Comparison of nutrient intake between cases and controls at baseline.** Truncated violin plots illustrate the distribution of nutrient intake for individual nutrients among individual cases (left) and controls (right). Each subplot represents a specific nutrient. The width of each violin plot corresponds to the density of observations at different intake levels. Central red dots represent the median, thick vertical red lines show the interquartile range, and thin lines (whiskers) indicate the range. Notable differences are observed between groups for some nutrients, reflecting variations in dietary adequacy. Statistical comparisons were conducted using Kruskal-Wallis to assess significance between groups.

Fig 4 compares the trends in mean nutrient intake of cases at baseline and 4 weeks after treatment. Overall, there appeared to be a clear pattern of increased nutritional intake at week 4 compared to week 0. However, the results show no significant difference between week 0 and week 4, except for folic acid, which showed a significant difference between the two time points (p = 0.02).

## Comparison based on clinical forms

Fig 5 illustrates the comparison of nutrient intakes across various clinical forms. The median nutrient intake was predominantly higher in individuals with plantar lesions and lower in those with papilloma and ulcer. Overall, nutrient intake did not differ significantly between clinical forms, except for fat (p = 0.04) and selenium (p = 0.01). However, Dunn's post hoc test with multiple comparison adjustment found that in the pairwise comparisons, ulcer vs. plantar was statistically significant for selenium (0.01) but not fat (p-value 0.05) (S3 and S4 Tables).

## Nutritional status and treatment outcomes of yaws cases

Out of the 64 cases of yaws, 56 (88%) healed, and 8 (12%) were not healed at the week 4 follow-up visit. All 9 (100%) cases classified as severe thinness healed after treatment, while 13/14 (93%) of cases classified as thinness healed after treatment. Among cases with normal nutritional status, 34/41 (83%) experienced healing, and 7/41 (17%) had not healed. The difference in healing outcomes across nutritional categories was not statistically significant (p = 0.41) (Table 6).

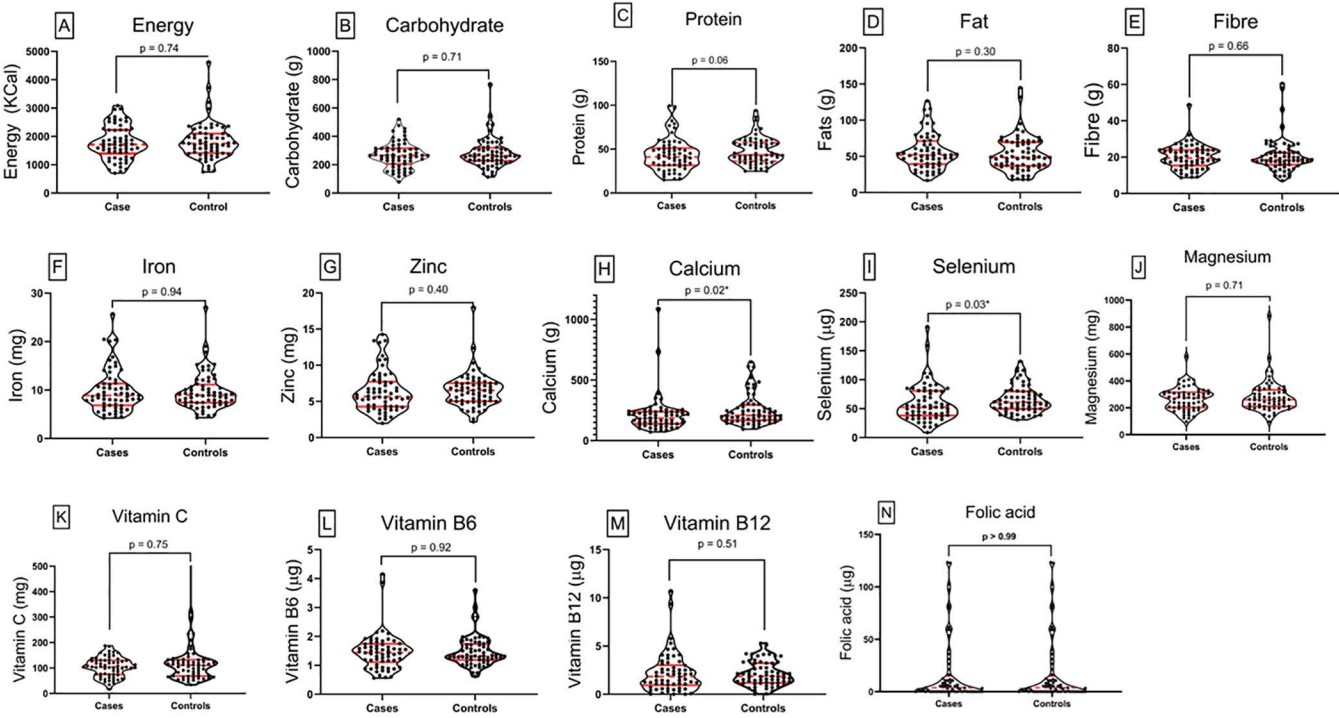

**Fig 3. Comparison of nutrient intake between cases and controls at follow-up.** Violin plots illustrate the distribution of nutrient intake for individual nutrients among individual cases (left) and controls (right). Each subplot represents a specific nutrient. The width of each violin plot corresponds to the density of observations at different intake levels. Central red dots represent the median, thick vertical red lines show the interquartile range, and thin lines (whiskers) indicate the range. Notable differences are observed between groups for some nutrients, reflecting variations in dietary adequacy, with a significant difference observed in selenium (p = 0.03). Statistical comparisons were conducted using Mann-Whitney's U-test to assess significance between groups.

Nutrient adequacy was evaluated to investigate possible links with healing outcomes (Table 7). No statistically significant differences were observed in nutrient adequacy between the healed and non-healed groups for all nutrients examined (p > 0.05 for all comparisons). However, some patterns stood out. A larger percentage of participants who did not heal had inadequate energy intake (88%, n = 7), compared to 80% in the healed group. Similar trends were seen for fibre (100% vs. 96%), iron (63% vs. 54%), zinc (75% vs. 71%), and vitamin B12 (63% vs. 55%). All participants, regardless of healing status, were deficient in calcium and folic acid. Adequate protein intake was slightly more frequent among non-healed participants (75%) than those who healed (66%).

The overall adequacy of vitamin C, vitamin B6, selenium, and magnesium was relatively high in both groups, with no substantial differences between healed and non-healed participants (Table 7).

## Discussion

To improve understanding of the role of nutrition on outcomes in yaws, we compared the nutritional status of yaws-affected children with their age- and sex-matched controls. Our findings offer valuable insights into the interplay between yaws and nutrition and allow for comparison with trends observed in other areas affected by NTDs. To our knowledge, this is the first study of its kind in a yaws-affected population, addressing the limited understanding of the nutritional status of individuals with yaws in Ghana.

The demographic characteristics of the study population were well-matched, minimizing confounding. Anthropometric assessments indicated concerning nutritional trends, with around a third (36%) of cases being classified as thin (14%

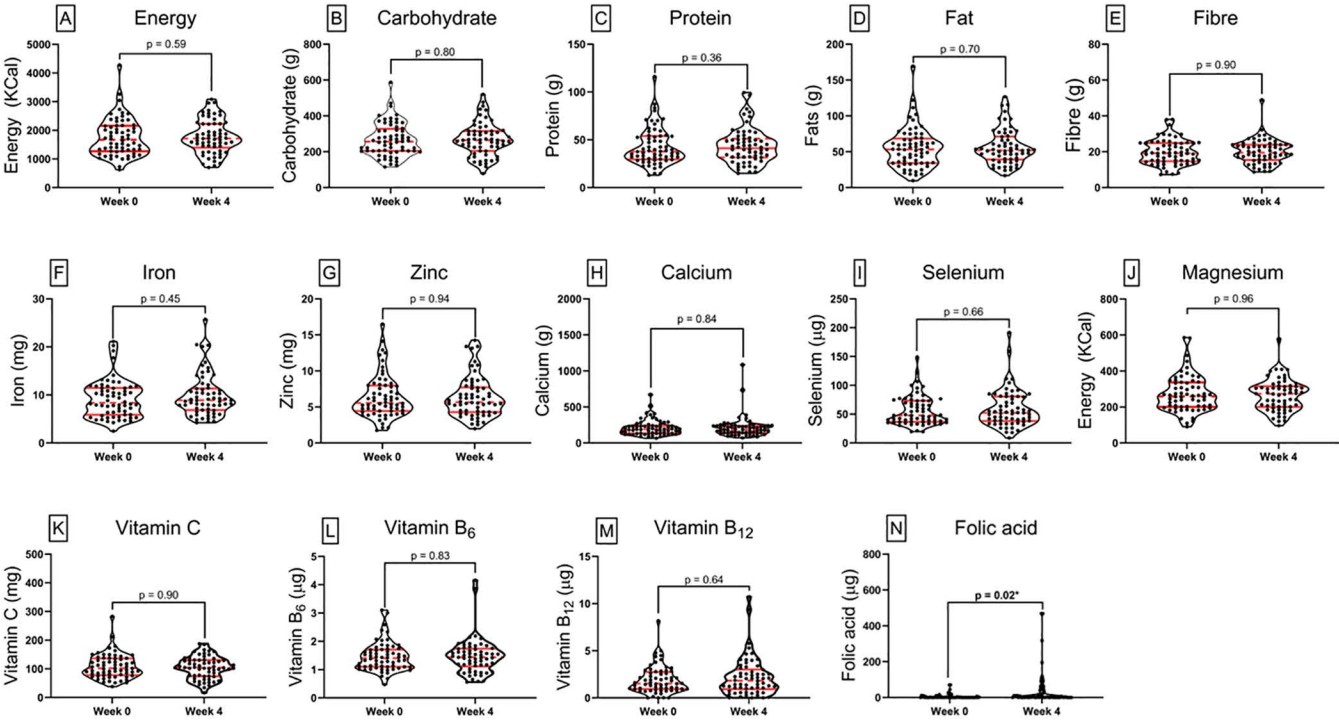

**Fig 4. Comparison of nutrient intake between cases at baseline and at follow-up.** Truncated violin plots illustrate the distribution of nutrient intake for individual nutrients among individual cases at baseline (week 0) and follow-up (week 4). Each subplot represents a specific nutrient. The width of each violin plot corresponds to the density of observations at different intake levels. Central red dots represent the median, thick vertical red lines show the interquartile range, and thin lines (whiskers) indicate the range. Notable differences are observed between groups for some nutrients, with a significant difference observed in folic acid (p = 0.02), reflecting variations in dietary adequacy. Statistical comparisons were conducted using Mann-Whitney's U-test to assess significance between groups.

severe thinness and 22% thinness) and 19% of controls classified as thin (6% severe thinness and 12% thinness). Although this difference was not statistically significant (p = 0.12), it highlights a high prevalence of undernutrition in this area, particularly in cases, possibly reflecting variabilities in dietary intake, the effect of exposure to infections, or socio-economic disadvantages in this population [50]. This is of public health concern, and addressing these disparities will require targeted interventions focusing on diet quality, infection prevention, and socioeconomic support to reduce growth faltering among this vulnerable group [51]. Comparable thinness rates have been documented in other parts of Ghana, including 30% in Tamale [52] in the north and 21.5% in Nkwanta [53] in the south. In those settings, thinness in children of school-going age was also linked to dietary patterns, infectious diseases [52], and the educational status of household heads [53].

Controls had a higher proportion of non-stunted individuals (73%, n = 47) and lower rates of moderate (22%, n = 14) and severe stunting (5%), compared to cases (67%, n = 43 not stunted; 27%, n = 17 moderate stunting; 6%, n = 4 severe stunting). Though these differences did not reach significance, it suggests that while chronic malnutrition is prevalent in the area, individuals infected with yaws may experience slightly greater nutritional impact.

Comparable stunting rates have been observed in populations affected by other NTDs, highlighting a consistent pattern of nutritional deficiencies. Studies on *Schistosoma mansoni* infection reported a stunting prevalence of 22.1% [54] but in our study, the proportion of both moderately and severely stunted may have been higher, with 33% (n = 21) of cases and 27% (n = 17) of controls affected. Similarly, in 2014, Diro et al. found that 63% of children under five and 50.5% of older

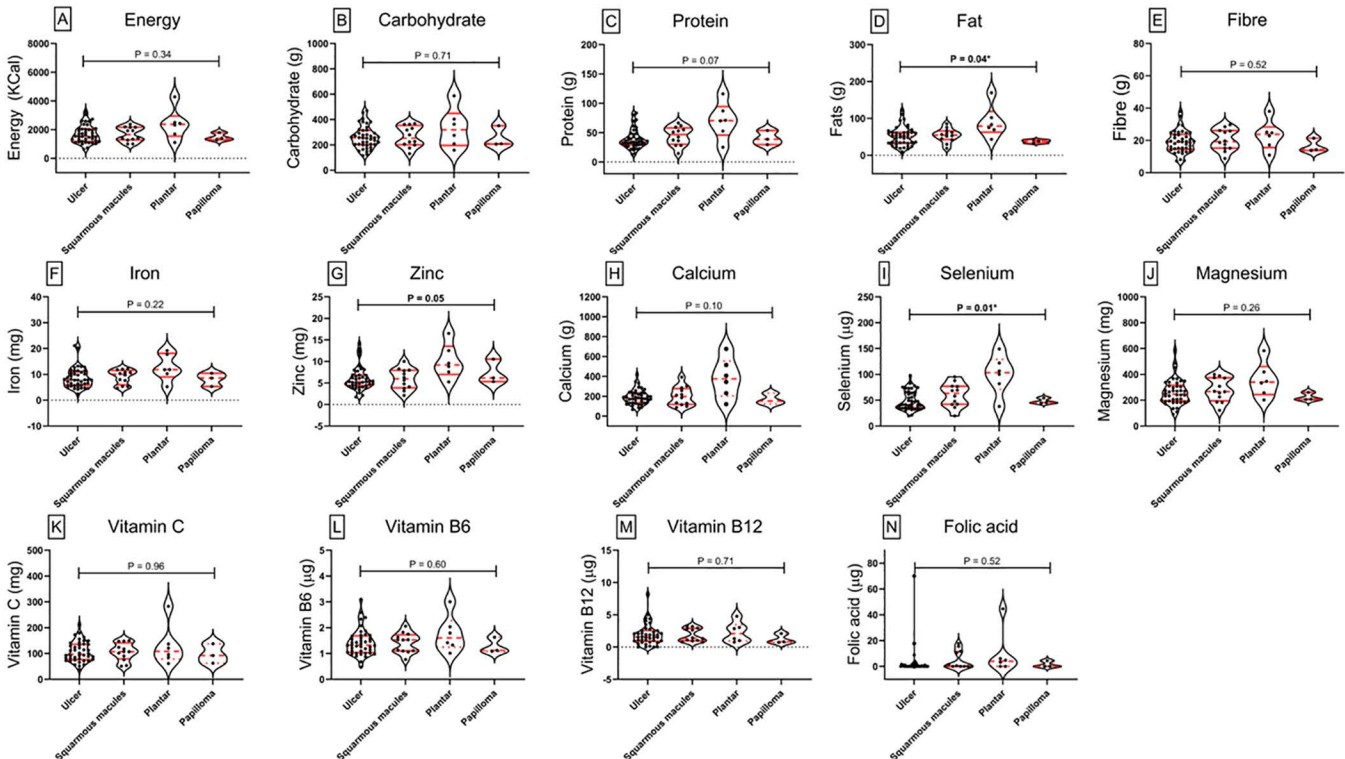

**Fig 5. Comparison of nutrient intake across lesion types.** Median, interquartile range comparison of lesion forms to nutrient intake was analyzed using the Kruskal-Wallis for non-parametric data for comparison between lesions, ulcer, papilloma, squamous macules, and plantar lesions, using the GraphPad Prism version 8 software (GraphPad Software Inc., USA). Each subplot illustrates the distribution of nutrient intake for individual nutrients across the lesion types at baseline, represented in a truncated violin plot.

**Table 6. Nutritional status of cases and treatment outcome.**

| Nutritional Status | Healed N = 56 (%) | Not healed N = 8 (%) | Overall N = 64 (%) | p value |
|---|---|---|---|---|
| Severe thinness | 9 (16) | 0 | 9 (14) | 0.41 |
| Thinness | 13 (23) | 1 (13) | 14 (22) | |
| Normal | 34 (61) | 7 (87) | 41 (64) | |

*Counts of nutritional status based on z-scores from WHO Anthroplus were considered statistically significant if p-value < 0.05 analyzed via Fisher's exact test. Anthropometric measure used BMI-for-age for study participants aged 5–19 [37].*

children affected by visceral leishmaniasis were stunted [55]. More recently, a study reported that nearly one-third (31%) of Buruli ulcer cases in a Ghanaian population were stunted [24]. Collectively, these findings reinforce the well-established association between NTDs and high stunting prevalence among affected populations [20,56–59].

Analysis of dietary intake revealed that both cases and controls consumed similar patterns of food groups. All participants universally consumed grains, roots, and tubers—an expected result, given that the Ghanaian diet is traditionally high in carbohydrate-rich foods made from starchy staples [60]. This aligns with findings from other regions where these staples are the dietary mainstay because of their easy availability and low cost [61].

Flesh foods were also eaten, with 77% of both cases and controls reporting consumption. However, only 68% of participants had enough protein according to quantification. Similarly, most participants (89%) consumed legumes, pulses,

**Table 7. Comparison of healing outcomes across nutrients.**

| Characteristic | Healed | Not Healed | Overall | p-value |
|---|---|---|---|---|
| | N = 56 (%) | N = 8 (%) | N = 64 (%) | |
| **Energy (Kcal)** | | | | >0.99 |
| Adequate | 11 (20) | 1 (13) | 12 (19) | |
| Inadequate | 45 (80) | 7 (88) | 52 (81) | |
| **Carbohydrates (g)** | | | | >0.99 |
| Adequate | 54 (96) | 8 (100) | 62 (97) | |
| Inadequate | 2 (4) | 0 (0) | 2 (3) | |
| **Protein (g)** | | | | >0.99 |
| Adequate | 37 (66) | 6 (75) | 43 (67) | |
| Inadequate | 19 (34) | 2 (25) | 21 (33) | |
| **Fibre (g)** | | | | >0.99 |
| Adequate | 2 (4) | 0 (0) | 2 (3) | |
| Inadequate | 54 (96) | 8 (100) | 62 (97) | |
| **Iron (mg)** | | | | 0.72 |
| Adequate | 26 (46) | 3 (38) | 29 (45) | |
| Inadequate | 30 (54) | 5 (63) | 35 (55) | |
| **Zinc (mg)** | | | | >0.99 |
| Adequate | 16 (29) | 2 (25) | 18 (28) | |
| Inadequate | 40 (71) | 6 (75) | 46 (72) | |
| **Selenium (µg)** | | | | >0.99 |
| Adequate | 44 (79) | 7 (88) | 51 (80) | |
| Inadequate | 12 (21) | 1 (13) | 13 (20) | |
| **Vitamin B12 (µg)** | | | | >0.99 |
| Adequate | 25 (45) | 3 (38) | 28 (44) | |
| Inadequate | 31 (55) | 5 (63) | 36 (56) | |
| **Vitamin C (mg)** | | | | >0.99 |
| Adequate | 54 (96) | 8 (100) | 62 (97) | |
| Inadequate | 2 (3.6) | 0 (0) | 2 (3.1) | |
| **Vitamin B6 (µg)** | | | | >0.99 |
| Adequate | 50 (89) | 8 (100) | 58 (91) | |
| Inadequate | 6 (11) | 0 (0) | 6 (9.4) | |
| **Magnesium (mg)** | | | | 0.70 |
| Adequate | 34 (61) | 6 (75) | 40 (63) | |
| Inadequate | 22 (39) | 2 (25) | 24 (38) | |
| **Calcium (mg)** | | | | >0.99 |
| Adequate | 0 (0) | 0 (0) | 0 (0) | |
| Inadequate | 56 (100) | 8 (100) | 64 (100) | |
| **Folic acid (µg)** | | | | >0.99 |
| Adequate | 0 (0) | 0 (0) | 0 (0) | |
| Inadequate | 56 (100) | 8 (100) | 64 (100) | |

*Proportions of cases categorized as healed and not healed at follow-up with adequate and inadequate energy and nutrient intakes were compared using a two-tailed Fisher's exact test using GraphPad Prism version 8. Categorical data are presented as percentages. Fisher's exact p-value (p < 0.05). Mean nutrient intake was categorized as adequate or inadequate based on the EAR and sometimes RDA using WHO/FAO values. The EAR and RDA cutoffs are based on age and sex [49].*

and nuts, with slightly more among cases (92%) than controls (86%). The most consumed in this group were cowpea and groundnut. Oil seeds, mainly palm nut (oil) and palm kernel oil, were the most frequently eaten (66% overall; 67% of cases and 64% of controls). This is expected, as oil palm is a major crop cultivated by local farmers in this area [31,62], making it accessible, affordable, and a key part of local culinary traditions.

However, intake of other nutrient-dense food groups was considerably lower. Only 35% of participants consumed green leafy vegetables, which are important sources of iron, calcium, and other micronutrients. Notably, the consumption was similar between cases and controls, suggesting generally low intake across the population.

Alarmingly, the consumption of fruit and vegetables was strikingly low (5% overall), with only 3% of cases and 8% of controls reporting fruit intake, and 8% (13% controls and 3% cases) consuming other vegetables. This highlights a significant gap in vitamin-rich food consumption and dietary fiber. Considering that fruits and vegetables are seasonal and often costly, research from other Ghanaian regions revealed increased consumption during bumper harvest seasons [63].

Similarly, dairy product consumption was limited to just 18% of the participants, with slightly higher intake among controls (22%) than cases (14%). This result, although modest, is not surprising, given that dairy consumption in Ghana is widely acknowledged as extremely low [64], possibly due to economic constraints or limited access to dairy products in the study area.

Egg consumption was also relatively low at 23%, indicating limited use of this accessible, nutrient-dense food. Despite accessibility and the numerous nutritional and health benefits that eggs present, and the long-term benefits of greatly minimizing acute malnutrition, their consumption has erroneously been linked to high cholesterol and cardiovascular diseases [64,65]. Additional reasons for low consumption of eggs in Ghana may be the lack of nutritional knowledge and cultural beliefs [66,67] although these were not assessed in the present study. Randomized controlled trials in Ghana and Ecuador reported that children on egg supplementation had a much higher increase in vitamin A [67] and a steady increase in the length-per-age, preventing stunting [68] compared to controls.

Our results reveal a pattern of dietary diversity primarily focused on starchy staples and protein sources, with lower consumption of fruits, vegetables, and dairy products. This imbalance may lead to micronutrient deficiencies and impact growth in children. It is important to implement interventions that enhance dietary diversity and encourage the inclusion of overlooked food groups, especially fruits, vegetables, eggs, and dairy. Policymakers and health promotion services can utilize existing platforms like the Information Service Department and thriving community information centres [69], to deliver informal education on nutrition, especially in rural Ghana. Additionally, information may be provided through the School Health Education Programme run by the Directorates of Education in the affected communities.

One of the most striking findings was the widespread lack of essential nutrients among the study participants. Whether participants had yaws or not, they faced energy inadequacies and some micronutrient deficits. An alarming finding was the universal inadequacy of calcium and folic acid (100% inadequacy) and the inadequate energy and fibre intake, with over 80% and 90% of the study population, respectively, failing to meet recommended levels. Protein inadequacy affected nearly a third of participants, and more than half had insufficient iron and zinc intake. Notably, deficiencies in iron, zinc, vitamin B12, and vitamin B6 were alarmingly common. These deficiencies with no significant difference between yaws cases and controls (although controls exhibited slightly higher intake in some nutrients), suggest that the nutritional challenges are systemic and may not be solely attributable to yaws infection. This mirrors the nutritional challenges observed in Buruli ulcer-endemic areas in Ghana [24], reinforcing the idea that malnutrition is not just a secondary concern but central to the health landscape in these communities. The lack of significant differences suggests that environmental and socioeconomic factors may play dominant roles in shaping dietary patterns.

Chronic undernutrition and micronutrient deficiencies are known to impair immune function, increase susceptibility to infections, and hinder physical and cognitive development [70,71]. Some studies have reported a strong association between micronutrient deficiency, infection, and immune response [72–74]. While yaws infection did not appear to worsen nutritional status compared to controls, the overall rates of malnutrition and micronutrient deficiencies are alarming and likely contribute to increased vulnerability to infectious diseases, including yaws.

Comparative studies in other NTDs, report similar deficiencies [20,56–59].

Interestingly, the comparison of nutrient intake across different clinical forms of yaws in this study revealed some variation, with the plantar group generally having higher median intake. Significant differences were observed in the intake of fat, and selenium. While the clinical significance of these differences is unclear, secondary yaws lesion is often accompanied by generalized lymphadenopathy and constitutional symptoms [75], and these could influence the diet provided by guardians. Additionally, these participants may have had greater access to animal-source foods rich in selenium due to their household income, availability, or guardian's occupation. Variability in dietary patterns may also be influenced by cultural preferences or seasonal food availability.

The majority of yaws cases responded well to treatment, with no significant association between nutritional status (as measured by BMI) and healing outcomes. Although a higher proportion of underweight children healed compared to those with normal BMI, this difference was not statistically significant. This finding contrasts with some studies on leprosy [76,77], and leishmaniasis [78] that reported that malnutrition may lead to a worse disease prognosis.

A greater proportion of participants who did not heal at follow-up had inadequate intake of energy, fibre, iron, zinc, and Vitamin B12, which are undoubtedly essential in tissue repair and the healing process [79]. Notably, all eight individuals who had unhealed lesions were deficient in both calcium and folic acid. These trends, although not statistically significant, may point to a cumulative effect of multiple mild-to-moderate deficiencies rather than a single nutrient deficiency driving poor outcomes.

Our study had some limitations. Reliance on 24-hour dietary recall may be biased and not fully reflect habitual intake. To enhance accuracy, recalls were done for typical days, avoiding festive seasons and atypical days, which would not have given a true reflection of dietary habits. No biochemical analysis was done to objectively assess nutritional deficiencies. Socioeconomic status and income levels of participants were not evaluated for their influence on dietary preferences. Furthermore, the sample size, while adequate for detecting moderate differences, may not have been sufficient to identify smaller yet clinically relevant differences. The study was undertaken in districts located in forested, food growing areas of Ghana and have two cropping seasons which define the annual food availability cycle [80]. Food availability is typically lower during the pre-harvest period (January- March) as food stored from the previous harvest depletes. The period of data collection coincided with the main farming season when food is generally available in these areas. We therefore expected food availability to be generally more during the period of the study. Yet, it is possible food availability at different periods in the year may have impacted nutrient intake of participants. Despite these limitations, our study provides valuable insights into the nutritional status of individuals with yaws in the study area and other similar settings.

The findings underscore the pressing need for comprehensive and integrated public health strategies to tackle both infectious diseases and nutritional deficiencies in yaws-endemic regions. Initiatives such as school-based nutrition programmes, community education, and improved access to a variety of nutrient-rich foods could help mitigate the high burden of malnutrition observed. Although the benefits may vary across affected children, such nutritional support could prove advantageous in these nutrient-deficient regions, potentially supporting weight gain, growth, and treatment outcomes [81]. To achieve a lasting impact on populations affected by yaws, MDA alone may not be sufficient; addressing malnutrition largely as a result of poverty in these areas may be critical [21].

## Conclusion

We found that individuals in these yaws-endemic districts face significant nutritional challenges, with high rates of undernutrition and micronutrient deficiencies observed in both yaws cases and controls. While nutritional status was not independently associated with worse treatment outcomes among yaws cases, the overall burden of malnutrition highlights the need for comprehensive health strategies in these communities to address the health challenges.

Further studies are needed in yaws endemic populations to ascertain the relationship between chronic nutritional deficiencies and clinical outcomes in yaws.

## Supporting information

**S1 Table. Distribution of intake diversity by sex among participants.**
(DOCX)

**S2 Table. Adequacy of nutrient intake by sex among participants.**
(DOCX)

**S3 Table. Dunn's multiple comparison for statistically significant micronutrient (Fat) within lesion types.**
(DOCX)

**S4 Table. Dunn's multiple comparison for statistically significant micronutrient (Selenium) within lesion type.**
(DOCX)

## Acknowledgments

We would like to sincerely thank the District Directors of Health Services and Education Service, Disease Control Officers, Community Leaders, Teachers, and pupils of Wassa Amenfi East and Aowin Districts for their collaboration. We want to thank Mr. Clement Tettey, Mr. Joseph Azabire, Mr. Owusu Boakye Yiadom, and other health staff of the study sites. We also thank Mr. Victor Yaw Morgan and Mr. Maxwel Adoko, and the Skin NTDs Research group at KCCR.

## Author contributions

**Conceptualization:** Abigail Agbanyo, Alex Owusu-Ofori, Richard Odame Phillips, Yaw Ampem Amoako.

**Data curation:** Abigail Agbanyo, Michael Ntiamoah Oppong.

**Formal analysis:** Abigail Agbanyo, Yaw Ampem Amoako.

**Investigation:** Abigail Agbanyo, Michael Ntiamoah Oppong, Ruth Dede Tuwor, Pius Takyi, Felix Wireko, Philemon Boasiako Antwi, Dzifa Kofi Ahiatrogah.

**Methodology:** Abigail Agbanyo, Alex Owusu-Ofori, Richard Odame Phillips, Yaw Ampem Amoako.

**Project administration:** Yaw Ampem Amoako.

**Resources:** Richard Odame Phillips, Yaw Ampem Amoako.

**Supervision:** Alex Owusu-Ofori, Richard Odame Phillips, Yaw Ampem Amoako.

**Validation:** Michael Ntiamoah Oppong, Dzifa Kofi Ahiatrogah, Aloysius Dzigbordi Loglo, Bernadette Agbavor, Alex Owusu-Ofori, Richard Odame Phillips.

**Visualization:** Michael Ntiamoah Oppong, Ruth Dede Tuwor, Pius Takyi, Felix Wireko, Philemon Boasiako Antwi, Dzifa Kofi Ahiatrogah, Aloysius Dzigbordi Loglo, Bernadette Agbavor.

**Writing – original draft:** Abigail Agbanyo, Yaw Ampem Amoako.

**Writing – review & editing:** Abigail Agbanyo, Michael Ntiamoah Oppong, Ruth Dede Tuwor, Pius Takyi, Felix Wireko, Philemon Boasiako Antwi, Dzifa Kofi Ahiatrogah, Aloysius Dzigbordi Loglo, Bernadette Agbavor, Alex Owusu-Ofori, Richard Odame Phillips, Yaw Ampem Amoako.

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
