## [Decision Letter · Decision Letter 0]

17 Sep 2025

Dear Dr. Amoako,

Thank you for submitting your manuscript to PLOS ONE. After careful consideration, we feel that it has merit but does not fully meet PLOS ONE’s publication criteria as it currently stands. Therefore, we invite you to submit a revised version of the manuscript that addresses the points raised during the review process.

We look forward to receiving your revised manuscript.

Kind regards,

Saki Raheem, PhD

Academic Editor

PLOS ONE

**Journal requirements:**

1. When submitting your revision, we need you to address these additional requirements. Please ensure that your manuscript meets PLOS ONE's style requirements, including those for file naming. The PLOS ONE style templates can be found at https://journals.plos.org/plosone/s/file?id=wjVg/PLOSOne_formatting_sample_main_body.pdf and https://journals.plos.org/plosone/s/file?id=ba62/PLOSOne_formatting_sample_title_authors_affiliations.pdf 2. We note that you have referenced “Abigail Agbanyo, Michael Ntiamoah Oppong, Dzifa Kofi Ahiatrogah, Ruth Dede Tuwor, Clement Tettey, Joseph Azabire, et al. and Abigail Agbanyo, Michael Ntiamoah Oppong, Ruth Dede Tuwor, Shadrach Mintah, Victor Yaw Morgan, Clement Tettey, et al.” which has currently not yet been accepted for publication. Please remove this from your References and amend this to state in the body of your manuscript: (Abigail Agbanyo, Michael Ntiamoah Oppong, Dzifa Kofi Ahiatrogah, Ruth Dede Tuwor, Clement Tettey, Joseph Azabire, et al. and Abigail Agbanyo, Michael Ntiamoah Oppong, Ruth Dede Tuwor, Shadrach Mintah, Victor Yaw Morgan, Clement Tettey, et al. [Submitted ]) as detailed online in our guide for authorshttp://journals.plos.org/plosone/s/submission-guidelines#loc-reference-style 3. Please upload a new copy of Figures 1 to 4 as the detail is not clear. Please follow the link for more information: https://blogs.plos.org/plos/2019/06/looking-good-tips-for-creating-your-plos-figures-graphics/" https://blogs.plos.org/plos/2019/06/looking-good-tips-for-creating-your-plos-figures-graphics/ 4. We note that there is identifying data in the Supporting Information file. Due to the inclusion of these potentially identifying data, we have removed this file from your file inventory. Prior to sharing human research participant data, authors should consult with an ethics committee to ensure data are shared in accordance with participant consent and all applicable local laws. Data sharing should never compromise participant privacy. It is therefore not appropriate to publicly share personally identifiable data on human research participants. The following are examples of data that should not be shared: -Name, initials, physical address-Ages more specific than whole numbers-Internet protocol (IP) address-Specific dates (birth dates, death dates, examination dates, etc.)-Contact information such as phone number or email address-Location data-ID numbers that seem specific (long numbers, include initials, titled “Hospital ID”) rather than random (small numbers in numerical order) Data that are not directly identifying may also be inappropriate to share, as in combination they can become identifying. For example, data collected from a small group of participants, vulnerable populations, or private groups should not be shared if they involve indirect identifiers (such as sex, ethnicity, location, etc.) that may risk the identification of study participants. Additional guidance on preparing raw data for publication can be found in our Data Policy (https://journals.plos.org/plosone/s/data-availability#loc-human-research-participant-data-and-other-sensitive-data) and in the following article: http://www.bmj.com/content/340/bmj.c181.long. Please remove or anonymize all personal information (<specific identifying information in file to be removed>), ensure that the data shared are in accordance with participant consent, and re-upload a fully anonymized data set. Please note that spreadsheet columns with personal information must be removed and not hidden as all hidden columns will appear in the published file. 5. Please include captions for your Supporting Information files at the end of your manuscript, and update any in-text citations to match accordingly. Please see our Supporting Information guidelines for more information: http://journals.plos.org/plosone/s/supporting-information.

Reviewers' comments:

Reviewer's Responses to Questions

**Comments to the Author**

1. Is the manuscript technically sound, and do the data support the conclusions?

Reviewer #1: Yes

Reviewer #2: Yes

2. Has the statistical analysis been performed appropriately and rigorously?

Reviewer #1: Yes

Reviewer #2: Yes

3. Have the authors made all data underlying the findings in their manuscript fully available?

Reviewer #1: Yes

Reviewer #2: Yes

4. Is the manuscript presented in an intelligible fashion and written in standard English?

Reviewer #1: Yes

Reviewer #2: Yes

**Reviewer #1:**  The manuscript “Dietary variability and micronutrient status of individuals with Yaws infection in Ghana: a case-control study” by Abigail Agbanyo et al. aims to reveal possible connections between nutritional status of individuals and yaws disease. This manuscript is well written and the topic is very interesting, but I have several concerns:

- Study setting:

o What was the reasoning behind selection of the studied communities and districts? Were they selected only because of consistent reports of yaws cases in the recent years, or were there other factors considered? Are these the only areas where cases of yaws are reported? I would appreciate more details and maybe a map of Ghana with the studied areas marked, as well as all the regions with yaws cases present.

- Study design:

o The participants were recruited over the period of seven months. Could the different times of the year have an influence on the nutrient intake? Discussion briefly mentions the seasonality of fruits and vegetables, does the season also affect availability of other food groups?

- Data analysis:

o Was any correction for multiple testing applied to the statistical tests used in the study?

- Results:

o In demographic characteristics description on lines 219 and 220, specify what is compared. It states that the sex distribution was not statistically significant, but the first part of the sentence mentions only 74% of males versus 26% females, which would be statistically significant difference.

o For the analysis of anthropometric characteristics, what data was used for the calculations? Baseline? Follow up? Both? Throughout the results, this information is stated only in some places. Please make this clear in both Methods and Results.

o Results are not divided by sex. It would be interesting to see such analysis as it would provide more details on how the nutrient intake differs between both sexes and if the nutrient intake inadequacy is more common in males or females. I only see the sex factor taken into account in the Adequacy of nutrient intake part of the Results, but not in the general analysis of the Nutrient intake and diversity. Is there a reason why this was not done?

o Figure 4 shows p-values, but which categories (clinical forms) were compared? The Dunn’s post-hoc test should identify which comparisons were statistically significant. Do the values shown always represent the comparison between Ulcers and Papillomas, as is currently marked in the Figure? Please add more details in the text.

- Discussion

o In general, the Discussion could be shorter. While the connections between different diseases and the stunting rates found in other studies and the possible effect of disease on nutrient intake or vice versa are very interesting, I would try to make this more concise.

Minor comments:

- Check the numbering of references. The citation on line 116 is numbered out of order.

- Line 163 and 166: Incorrect term. Sex should be used instead of gender.

- Text on Figures is too small and almost unreadable.

**Reviewer #2:**  This is a nice negative study demonstrating that nutrition seems to have little/no role in yaws but that under-nutrition is commonly found in yaws endemic communities.

Major Comments:

The methods are fairly well describe but there isnt a sample size section - was this calculated or was the sample size based on convenience of matching cases found in the previous study to controls? If there was a sample size calculation if should be included.

Please give a bit more detail on the matching of controls. For example I would have considered a similar age/gender might be important or potentially selecting someone without yaws but within the same household which would provide additional control for variations in exposure.

I would be more cautious in saying there are micronutrient differences - as the authors note the differences are not statistically significant and even if they were I dont think these are clinically significant differences i.e a <1% difference in Vitamin C intake etc.

There is a lot of testing (i,e if each micronutrient is associated with lesion type) - this is at least 13 comparisons; I would consider adjusting for multiple comparisons which i suspect would make even the weak associations detected become non-significant. (Similar point for table 7)

I suspect the discussion could be tigheter and shorter given the essentially null findings.

Minor Comments:

Drug names like Azithromycin dont need to be capitalised.

**Do you want your identity to be public for this peer review?** For information about this choice, including consent withdrawal, please see our Privacy Policy

Reviewer #1: No

Reviewer #2: No

---

## [Author Response · Author response to Decision Letter 1]

26 Sep 2025

26th September 2025

Kumasi Centre for Collaborative Research

Kwame Nkrumah University of Science and Technology

Kumasi, Ghana

The Editor

PLOS One Journal

Dear Sir,

Re: ‘Dietary variability and micronutrient status of individuals with Yaws infection in Ghana: a case-control study’ [PONE-D-25-45732]

My co-authors and I have taken note of the review comments and revised the manuscript as suggested.

We wish to submit the revised manuscript for publication in your esteemed journal. In the attached reply, the reviewer questions are in blue font and our responses are in red font.

We look forward to the next steps towards the publication of our manuscript.

Yours Sincerely,

Dr Yaw Ampem Amoako

Corresponding author

Point-by-point response to review comments

Response: Done as required.

2. We note that you have referenced “Abigail Agbanyo, Michael Ntiamoah Oppong, Dzifa Kofi Ahiatrogah, Ruth Dede Tuwor, Clement Tettey, Joseph Azabire, et al. and Abigail Agbanyo, Michael Ntiamoah Oppong, Ruth Dede Tuwor, Shadrach Mintah, Victor Yaw Morgan, Clement Tettey, et al.” which has currently not yet been accepted for publication. Please remove this from your References and amend this to state in the body of your manuscript: (Abigail Agbanyo, Michael Ntiamoah Oppong, Dzifa Kofi Ahiatrogah, Ruth Dede Tuwor, Clement Tettey, Joseph Azabire, et al. and Abigail Agbanyo, Michael Ntiamoah Oppong, Ruth Dede Tuwor, Shadrach Mintah, Victor Yaw Morgan, Clement Tettey, et al. [Submitted ]) as detailed online in our guide for authors

Response: Done as suggested. We have deleted those references from the manuscript in line with the journal recommendations.

3. Please upload a new copy of Figures 1 to 4 as the detail is not clear. Please follow the link for more information: https://blogs.plos.org/plos/2019/06/looking-good-tips-for-creating-your-plos-figures-graphics/" https://blogs.plos.org/plos/2019/06/looking-good-tips-for-creating-your-plos-figures-graphics/

Response: The figures have been reformatted as suggested.

4. We note that there is identifying data in the Supporting Information file. Due to the inclusion of these potentially identifying data, we have removed this file from your file inventory. Prior to sharing human research participant data, authors should consult with an ethics committee to ensure data are shared in accordance with participant consent and all applicable local laws.

-Location data

Response: We have removed S1 from the submission. We have included a reference to the skin diseases reporting and recording form (line 157).

Response: We have opted to provide a reference to the skin diseases reporting and recording form (line 157). The S1 file has therefore been deleted from the submission.

Reviewer #1: The manuscript “Dietary variability and micronutrient status of individuals with Yaws infection in Ghana: a case-control study” by Abigail Agbanyo et al. aims to reveal possible connections between nutritional status of individuals and yaws disease. This manuscript is well written and the topic is very interesting, but I have several concerns:

- Study setting:

o What was the reasoning behind selection of the studied communities and districts? Were they selected only because of consistent reports of yaws cases in the recent years, or were there other factors considered? Are these the only areas where cases of yaws are reported? I would appreciate more details and maybe a map of Ghana with the studied areas marked, as well as all the regions with yaws cases present.

Response: As indicated in the manuscript (line 125-127), the districts were selected based on their consistent report of yaws cases. Furthermore, the skin NTD Research group collaborates with the Ghana Health Service to operate clinics for individuals with skin NTDs including yaws in these districts. We have included a map of Ghana with the study districts as figure 1 in line with the review recommendation.

- Study design:

o The participants were recruited over the period of seven months. Could the different times of the year have an influence on the nutrient intake? Discussion briefly mentions the seasonality of fruits and vegetables, does the season also affect availability of other food groups?

Response: We have added the following sentences to the limitations section (line 556-563) to address the review concern:

‘The study was undertaken in districts located in forested, food growing areas of Ghana and have two cropping seasons which define the annual food availability cycle. Food availability is typically lower during the pre-harvest period (January- March) as stored food from the previous harvest depletes. The period of data collection coincided with the main farming season when food is generally available in these areas. We therefore expected food availability to be generally more during the period of the study. Yet, it is possible food availability at different periods in the year may have impacted nutrient intake of participants.’

- Data analysis:

o Was any correction for multiple testing applied to the statistical tests used in the study?

Response: We indicated in the section on data analysis (line 219-222) that ‘Multiple comparisons employed Dunn's post-hoc correction in conjunction with the Kruskal-Wallis test. Since numerous nutrients correlated with one another, multiple t-tests were employed to compare the nutrient consumption of cases and controls.’

- Results:

o In demographic characteristics description on lines 219 and 220, specify what is compared. It states that the sex distribution was not statistically significant, but the first part of the sentence mentions only 74% of males versus 26% females, which would be statistically significant difference.

Response: Done as suggested. We have revised the section (line 230-231) to read:

‘there was no statistically significant difference (p > 0.99) in sex distribution between yaws cases and controls.’

o For the analysis of anthropometric characteristics, what data was used for the calculations? Baseline? Follow up? Both? Throughout the results, this information is stated only in some places. Please make this clear in both Methods and Results.

Response: We have added the following sentence to the methods section (line 209-210) to address this review comment:

‘In the current study, the baseline anthropometric measurements were used in the analysis of anthropometric characteristics.’

We have also revised the title of table 1 as follows:

‘Table 1 Demographic and anthropometric characteristics of study participants at baseline’

Results are not divided by sex. It would be interesting to see such analysis as it would provide more details on how the nutrient intake differs between both sexes and if the nutrient intake inadequacy is more common in males or females. I only see the sex factor taken into account in the Adequacy of nutrient intake part of the Results, but not in the general analysis of the Nutrient intake and diversity. Is there a reason why this was not done?

Response: Many thanks for the suggestion. We have included the analysis of food intake diversity and nutrient adequacy by sex of participants in the results (line 274-275 and 326-327) and as Supporting information (S1 Table and S2 Table respectively).

Figure 4 shows p-values, but which categories (clinical forms) were compared? The Dunn’s post-hoc test should identify which comparisons were statistically significant. Do the values shown always represent the comparison between Ulcers and Papillomas, as is currently marked in the Figure? Please add more details in the text.

Response: The reference to the Dunn’s post hoc test in the figure has been deleted. The results of the Dunn’s post hoc test are now presented in Supplementary Tables 3A and 3B.

- Discussion

In general, the Discussion could be shorter. While the connections between different diseases and the stunting rates found in other studies and the possible effect of disease on nutrient intake or vice versa are very interesting, I would try to make this more concise.

Response: WE have deleted some sentences to shorten the discussion section as suggested.

Minor comments:

- Check the numbering of references. The citation on line 116 is numbered out of order.

Response: This has been corrected.

- Line 163 and 166: Incorrect term. Sex should be used instead of gender.

Response: Done as suggested

- Text on Figures is too small and almost unreadable.

Response: The figures have been reformatted as suggested.

Reviewer #2: This is a nice negative study demonstrating that nutrition seems to have little/no role in yaws but that under-nutrition is commonly found in yaws endemic communities.

Response: Many thanks for the review comments.

Major Comments:

The methods are fairly well describe but there isnt a sample size section - was this calculated or was the sample size based on convenience of matching cases found in the previous study to controls? If there was a sample size calculation if should be included.

Response: Due to the focal endemicity and uneven distribution of yaws as a disease, no formal sample size calculation was done, and the sample size was based on the convenience of matching cases identified during the active case search to controls. This has been clarified in the methods section (line 150-152).

Please give a bit more detail on the matching of controls. For example I would have considered a similar age/gender might be important or potentially selecting someone without yaws but within the same household which would provide additional control for variations in exposure.

Response: We agree with the reviewer’s suggestion that selection of age/ sex matched individuals from within the same household would have provided additional control for variations in exposure. However, this was not always possible due to the absence of such controls in the household of study participants.

We have revised the section to indicate that:

‘Age and sex matched controls were selected from within the same household as cases. Where no age and sex matched controls were available from within the same household, individuals living in the same community but who had negative DPP test results and did not have any clinical evidence of yaws were selected as controls.’

I would be more cautious in saying there are micronutrient differences - as the authors note the differences are not statistically significant and even if they were I dont think these are clinically significant differences i.e a <1% difference in Vitamin C intake etc.

Response: We have deleted the word ‘severe’ from the text and the revised section now reads (line 510):

‘Whether participants had yaws or not, they faced energy inadequacies and some micronutrient deficits.’

There is a lot of testing (i,e if each micronutrient is associated with lesion type) - this is at least 13 comparisons; I would consider adjusting for multiple comparisons which i suspect would make even the weak associations detected become non-significant. (Similar point for table 7)

Response: We have applied Dunn’s post-hoc correction. These results have been included in the main manuscript text (line 386-388) and as S3 Table and S4 Table.

I suspect the discussion could be tigheter and shorter given the essentially null findings.

Response: We have deleted some sentences from the discussion section to shorten the manuscript.

Minor Comments:

Drug names like Azithromycin dont need to be capitalised.

Response: Revised as suggested.

---

## [Decision Letter · Decision Letter 1]

30 Sep 2025

Dietary variability and micronutrient status of individuals with Yaws infection in Ghana: a case-control study

PONE-D-25-45732R1

Dear Dr. Amoako,

We’re pleased to inform you that your manuscript has been judged scientifically suitable for publication and will be formally accepted for publication once it meets all outstanding technical requirements.

Kind regards,

Saki Raheem, PhD

Academic Editor

PLOS ONE

Additional Editor Comments (optional):

Reviewers' comments:

Reviewer's Responses to Questions

**Comments to the Author**

Reviewer #2: (No Response)

2. Is the manuscript technically sound, and do the data support the conclusions?

Reviewer #2: (No Response)

3. Has the statistical analysis been performed appropriately and rigorously?

Reviewer #2: (No Response)

4. Have the authors made all data underlying the findings in their manuscript fully available?

Reviewer #2: (No Response)

5. Is the manuscript presented in an intelligible fashion and written in standard English?

Reviewer #2: (No Response)

Reviewer #2: I am satisfied the authors have responded to the comments adn the paper can be accepted for publication.

**Do you want your identity to be public for this peer review?** For information about this choice, including consent withdrawal, please see our Privacy Policy

Reviewer #2: No

---

## [Editor Report · Acceptance letter]

PONE-D-25-45732R1

PLOS ONE

Dear Dr. Amoako,

I'm pleased to inform you that your manuscript has been deemed suitable for publication in PLOS ONE. Congratulations! Your manuscript is now being handed over to our production team.

Kind regards,

on behalf of

Dr. Saki Raheem

Academic Editor

PLOS ONE